# Bactericidal Effect of Underwater Plasma Treatment on Waste Brine from Kimchi Production Process and the Evaluation of Reusability of Plasma-Treated Waste Brine in Salting Kimchi Cabbage

**DOI:** 10.3390/foods12040728

**Published:** 2023-02-07

**Authors:** Junghyun Lim, Eun Jeong Hong, Seong Bong Kim, Mi-Ai Lee, Seungmin Ryu

**Affiliations:** 1Institute of Plasma Technology, Korea Institute of Fusion Energy, Gunsan-si 54004, Republic of Korea; 2Practical Technology Research Group, World Institute of Kimchi, Gwangju 61755, Republic of Korea

**Keywords:** underwater plasma, waste brine, reuse, disinfection, Kimchi cabbage

## Abstract

Recycling waste brine from the Kimchi production process is an important environmental and industry issue. We applied an underwater plasma to reduce food-borne pathogens in the waste brine. The capillary electrodes supplied by alternating current (AC) bi-polar pulsed power were applied to treat 100 L of waste brine. The inactivation efficacy was analyzed using four different agars (Tryptic Soy Agar (TSA), Marine agar (MA), de Man Rogosa Sharpe Agar (MRS), and Yeast Extract–Peptone–Dextrose (YPD), respectively. The microbial population was reduced linearly with treatment time, irrespective of culturing medium. The inactivation followed the log-linear model (R^2^: 0.96–0.99). The reusability of plasma-treated waste brine (PTWB) was determined by five parameters (salinity, pH, acidity, reducing sugar, and the population of microorganisms) of salted Kimchi cabbage, and they were compared with newly made brine (NMB) and waste brine (WB), respectively. The results showed that the quality of salted Kimchi cabbage prepared by PTWB was not significantly different from that of NMB, indicating that the underwater plasma treatment is feasible to reuse waste brine in the salting process of Kimchi production.

## 1. Introduction

Kimchi is one of the most famous traditional fermented food in Korea. The most popular ingredient of Kimchi is salted Kimchi (or napa) cabbage. Kimchi cabbage (*Brassica rapa* L. ssp. *pekinensis*) is a source of several phytochemicals including flavonoids, carotene, and ascorbic acid [1]. The consumer demand for commercially processed salted Kimchi cabbage is increased as it is convenient and easy to make Kimchi. Thus, the production of safe and healthy salted Kimchi cabbage is the most important requirement for consumers. Nevertheless, salted Kimchi cabbage can be contaminated by foodborne bacteria from various sources. As Kimchi cabbage is grown in soil, many soil-induced microorganisms are associated with the contamination of Kimchi cabbage [2]. In addition, salt, usually sun-dried salt, is also a source of bacteria. Numerous kinds of bacteria have been found in sun-dried salt including *Staphylococcus aureus*, *Halophilic bacteria*, and other aerobic bacteria and coliforms [3].

The production of salted Kimchi cabbage can be prepared with the following sequences: salting with about 10% brine, desalination with tap water or groundwater, and storage [4]. Thus, a large amount of brine is consumed in the manufacturing of salted Kimchi cabbage. There is no doubt that clean brine should be used in this process. However, spent brine which must contain several microorganisms is not discharged but is reused due to economic problems. Normally, the brine is reused three times in the summer, and five times in the winter season without proper sterilization processes [5]. As a result, in 2012, about 1200 students were infected with pathogenic *Escherichia coli* (*E. coli*) in Kimchi causing outbreaks. Much research has been conducted to reuse waste brine using ozone [6], sodium hypochlorite (NaOCl) [7], and pasteurization [8]. These treatments show high inactivation efficacy in a wide range of microorganisms. However, problems have been found with these methods including the health risks associated with the formation of potential carcinogens such as haloacetic acids and trihalomethanes, or long treatment times [9]. Therefore, developing an alternative sterilization process for the hygienic manufacture of salted Kimchi cabbage is needed. Additionally, sterilized waste brine should be examined for use in the salting process and compared to freshly made brine to test its reusability.

Plasma treatment has shown great potential in various applications, such as wastewater treatment, biomedical applications, and the food washing process due to its simplicity, effectivity for degrading organic materials, and environmentally friendly technology [10,11,12,13,14]. The use of plasma received increasing attention from the food industry due to its high efficiency in producing free radicals that do not leave any residual secondary pollutants. In particular, the application of underwater plasmas has gained considerable attention. Plasma generation in water produces short-lived species such as OH radicals and superoxide radicals that have a high oxidation potential of 2.8 eV and 1.0 eV, respectively [14]. These short-lived species can react with target materials directly resulting in the fast degradation of toxic contamination without a pH drop which is not generally preferred in the food industry. Moreover, it was reported that underwater plasma is more competent than plasma discharge in contact with water [15]. The underwater plasma treatment is effective not only to reduce the population of many kinds of microorganisms but also to decompose the colorant in chili powder, Rhodamine B [16]. We have investigated capillary discharge, one of the ways to generate underwater plasma, to eliminate pollutants for the last decade [17,18,19,20]. We have discovered that a metal electrode covered by a ceramic tube is an effective way to make underwater plasma, and a smaller electrode diameter (smaller than 3 mm) and a higher pulse width (longer than 1 μS) is better to produce oxidants [20]. We have also found that underwater capillary discharge is an effective way to eliminate microorganisms between 10 and 50 μm [17]. A novel capillary electrode combined with a grounded nozzle supporter has recently been found to generate plasma-activated spray which enables it to kill microorganisms suspended in the air, such as those found in greenhouses or food storage [21]. Despite these studies, underwater plasma in the food industry is not fully understood.

Therefore, in this study, we employed underwater plasma to reuse waste brine. We investigated the effect of the treatment time on the inactivation of different microorganisms in waste brine. We also applied plasma-treated waste brine (PTWB) in the salting process and the results were compared to waste brine (WB) and newly made brine (NMB) to determine its reusability.

## 2. Materials and Methods

### 2.1. Sample Preparation

The WB used in this experiment was supplied by a “K” factory located in Gwangju, Korea. Fresh Kimchi cabbage was purchased from the local market in Gwangju, Korea. The inner leaves, outer leaves, and bottom of Kimchi cabbage were removed, and the white ribs were cut into 3 × 3 cm.

### 2.2. Plasma Device and Operation

The schematic drawing of the underwater plasma reactor is presented in Figure 1. The reactor consists of cylindrical stainless steel with an outer diameter of 100 mm, an inner diameter of 90 mm, and a length of 500 mm. Eighteen capillary electrodes were inserted into the reactor (120-degree intervals, 3 pins in each of the 6 levels), and 6 viewports of quartz were installed to observe the discharge. The tungsten electrodes, with a diameter of 2 mm, were covered with a cylindrical aluminum oxide (Al_2_O_3_) dielectric capillary with an inner diameter of slightly larger than 2 mm, and an outer diameter of 4 mm as shown in Figure 1a. The distance between the tips of the tungsten electrode and the Al_2_O_3_ tube was fixed at 2 mm. The plasma was discharged by applying alternating current (AC) bi-polar pulsed power (AP150-02-02, EESYS, Seongnam, Republic of Korea). The treatment system is shown in Figure 1b. The The WB was stored in a 100 L tank. A water pump was used to maintain the flow rate of approximately 83 L min^−1^. The big particles in the WB were removed using filters, and their diameters were 5 and 1 mm. A total of 100 L of WB was treated by underwater plasma for various treatment times: 0, 10, 20, and 30 min. After each treatment, samples were obtained to measure the population of microorganisms using different agars. During the plasma discharge, applied voltage and current profile during plasma treatment were measured by a high voltage probe (P6015A Tektronix, Beaverton, OR, USA) and a fast-current monitor (#110 Pearson Electronics, Palo Alto, CA, USA), respectively. The monitored voltage and current signals were recorded by a digital oscilloscope (DPO2024, Tektronix, Beaverton, OR, USA). The optical emission spectrum (OES) was measured using a spectrometer (HR4000, Ocean Optics, Dunedin, FL, USA).

### 2.3. Water Analysis

The concentration of hydrogen peroxide was measured every 10 min using the titanium sulfate method [13]. The color difference was determined using a UV-Vis spectrophotometer (HS-3300, Humas, Daejeon, Republic of Korea) with the detection of UV absorbance at 405 nm. The chemical oxygen demand (COD) (HACH Method 8000) was measured using a spectrophotometer (DR 6000, HACH, Loveland, CO, USA) in accordance with the manufacturer’s instruction. Temperature and pH were measured using a multiparameter meter (HI9829, HANNA Instrument, Seoul, Republic of Korea) during plasma treatment after calibration.

### 2.4. Preparation of Salted Kimchi Cabbage

The salting and desalting processes were conducted to make salted Kimchi cabbage. For the salting process, 200 g of Kimchi cabbage samples were in 400 mL of salt solution (NMB, WB, or PTWB) for 1 h at room temperature. After that, the cabbage was immersed in 400 mL of tap water three times for one minute each for desalination. After the desalting process, bacterial counts and quality tests were conducted, respectively.

### 2.5. Bacterial Sample Analysis

A colony count assay was used for the analysis of the microbial population in WB. A sample of 1 mL was serially diluted in 0.85% saline solution (3M Korea, Seoul, Republic of Korea) for the bacterial count in WB. The appropriate dilutions were plated in triplicate on four types of media: Tryptic Soy Agar (TSA, Kisanbio, Seoul, Republic of Korea) was used for the total aerobic bacterial count, de Man Rogosa Sharpe Agar (MRS, Kisanbio, Seoul, Republic of Korea) was used for the counting of lactic acid bacteria, Marine agar (MA, Kisanbio, Seoul, Republic of Korea) was used for the counting of halophilic bacteria, and Yeast Extract–Peptone–Dextrose (YPD, Kisanbio, Seoul, Republic of Korea) was used for the counting of maintaining and propagating yeasts. The TSA, MA, MRS, and YPD plates were incubated at 30 °C for 48–72 h. Bacterial counts in WB were expressed as colony forming unit (CFU) per 1 mL. For a more detailed investigation, 16s rRNA gene sequencing of microorganisms isolated from the WB was performed as the previously introduced method [22]. In brief, the microorganism colony was cultured in TSA, MRS, MA, and YPD. After cultivation, the DNA extraction was performed using a PowerSoil DNA Isolation Kit (MO BIO Laboratories, Carlsbad, CA, USA). Polymerase chain reaction (PCR) amplification was performed with extracted DNA and using primers targeting the V3 and V4 regions of the 16S rRNA gene. Amplifications were performed under the following conditions: initial denaturation at 95 °C for 3 min, followed by 30 cycles of denaturation at 95 °C for 20 s, primer annealing at 72 °C for 15 s, and extension at 72 °C for 60 s, with a final elongation step performed at 72 °C for 5 min. Sequencing was performed with the Illumina MiSeq platform by Macrogen (Macrogen Inc., Seoul, Republic of Korea). After 0, 10, 20, and 30 min of underwater plasma treatment, 1 mL of the sample was withdrawn from the 100 l of the saline tank. The population of each plasma-treated microorganism was measured using the same method as that of measuring the population of microorganisms in WB. For the bacterial count in the salted Kimchi cabbage, 10 g of Kimchi cabbage and 100 mL of 0.9% saline water were homogenized with a stomacher (Bagmixer R400, Interscience, Saint Nom, France) for 1 min, and the supernatant was placed onto 4 different agars (TSA, MA, MRS, and YPD). All agars were then incubated at 30 °C for 48–72 h following the manufacturer’s instructions. Bacterial counts for the salted Kimchi cabbage were expressed as colony forming units (CFUs) per 1 g of Kimchi cabbage.

### 2.6. Modeling of Bacterial Inactivation

To model the bacterial response under underwater plasma discharge, two frequently employed models were used [23,24]. The models used in this study were linear curves (log-linear) and non-linear curves (Weibull).

#### 2.6.1. Log-Linear Model

The log-linear model is the most frequently accepted model for predicting the inactivation patterns of bacteria. It assumes that bacterial inactivation shows first-order kinetics as follows [24]:(1)$$\log\frac{N_{t}}{N_{0}}=-kt$$
where $N_{0}$ and $N_{t}$ (CFU mL^−1^) represent the surviving populations of bacteria before treatment or after treatment time *t* (min), respectively. k represents kinetic constant (min^−1^).

#### 2.6.2. Weibull Inactivation Model

Despite the log-linear model being a widely accepted model, some microbial inactivation curves tend to exhibit non-log-linear behavior. Thus, the Weibull regression model was employed for analyzing non-log-linear inactivation patterns as follows [24]:(2)$$\log\frac{N_{t}}{N_{0}}=-\frac{1}{2.303}{\left(\frac{t}{\alpha }\right)}^{\beta }$$
where α represents characteristic time (min) and β represents the shape of the curve (unitless). When β = 1, the curve shows linear. Alternately, it could be a tailing (β<1) or a shoulder pattern (β>1).

### 2.7. Analysis Salinity, pH, Acidity, and Reducing Sugar Content

To determine the salinity, each Kimchi cabbage sample was homogenized using a blender (HR1372, Philips, Guildford, UK). The salinity of the homogenized sample was analyzed using Mohr’s titration method. Briefly, after the sample filtration with Whatman No. 2 filter paper, a mixture of 10 mL of the filtered sample solution and 1 ml of 2% potassium chromate indicator solution was titrated against 0.02 N AgNO_3_ until a red-brown color (at the endpoint) was obtained. The pH values of the homogenized sample were measured using pH meter (TitroLine Easy, SI Analytics, Mainz, Germany). The acidity was assessed by titrating to 10 mL of the filtrate by adding 0.1 N NaOH to reach the endpoint (pH 8.3). The consumed NaOH was calculated and converted to lactic acid content in % with following equation:$$\mathrm{Acidity}\;\left(\%\right)=\frac{\mathrm{consumed}\;0.1\;N\;\mathrm{NaOH}\;\left(\mathrm{mL}\right)\times \;\mathrm{factor}\;\mathrm{of}\;0.1\;N\;\mathrm{NaOH}\;\times 0.009}{\mathrm{Sample}\;\mathrm{solution}\;\left(\mathrm{mL}\right)}\times 100$$

For the determination of reducing sugar content, 1 mL of each diluted sample was mixed with 3 mL of 3, 5-dinitrosalicylic acid reagent in a glass test tube. The mixture was boiled at 100 °C for 5 min. After cooling to room temperature, the absorbance of the solution was determined at 550 nm. Reducing sugar contents (mg L^−1^) were expressed as glucose equivalents.

### 2.8. Statistical Analysis

SPSS statistics 24 software (SPSS Inc., Chicago, IL, USA) was employed for data analysis as previously used method [13]. Three independent experiments were conducted and the results were analyzed using the analysis of variance (ANOVA). The values from all experiments were expressed as the mean ± standard deviation. Duncan’s multiple range test was used to evaluate the significance of the differences (*p* < 0.05).

## 3. Results

### 3.1. Characteristics of Waste Brine

Table 1 presents the characteristics of WB from the salting process of Kimchi cabbage. The pH value of WB was slightly acidic, probably due to the presence of organic acid such as lactic acid [25]. The conductivity and salinity of NMB were approximately 140 mS cm^−1^ and 10% (data not shown), but some salts were absorbed into the Kimchi cabbage during the salting process and residual salinity was about 7%. Before the salting process, Kimchi cabbage was normally stored in a container for several weeks which may induce low freshness. This step can cause old leaves or organic matter in Kimchi cabbage, which increase in COD value during the salting process [26]. The bacterial population was measured using four different culturing agars (TSA for all aerobic bacteria, MA for counting halophilic bacteria, MRS for lactic acid bacteria, and YPD for the counting of maintaining and propagating yeasts). The results show that halophilic bacteria were the most concentrated, whereas a small amount of yeasts were found. For more investigation on living bacteria in WB, a 16s rRNA sequencing test was conducted and the results are presented in Appendix A. Briefly, nine *Bacillus* sp., three *Microbacterium* sp., two *Curtobacterium* sp., two *Sanguibacter* sp., two *Paenibacillus* sp., and two *Rhodococcus* sp. were found. In addition, one *Kocuria* sp., *Psychrobacter* sp., *Sphingomonas* sp., *Hypocreales* sp., *Arthrobacter* sp., *Staphylococcus* sp., *Marinilactibacillus* sp., *Plantibacter* sp., *Plectosphaerella* sp., *Aeromicrobium* sp., *Alternaria* sp., *Verticillium* sp., *Pantoea* sp., *Rathayibacter* sp., and *Lactococcus* sp., and *Serratia marcescens* were also identified.

### 3.2. Plasma Characteristics

Voltage and current signals during the underwater plasma discharge in 10% *w v*^−1^ brine are shown in Figure 2a. The peak voltage is 0.7 kV and the peak current is 30 A. The frequency is 20 kHz and the pulse width are 7.5 microseconds. The active power was calculated using V-I signals using Equation (3) and the result was about 4.5 kW [27].
(3)$$P(W)=\frac{1}{T}\int_{0}^{T}U(t)\times I(t)\times dt$$
where *P*(*W*) is the output power; *U*(*t*) and *I*(*t*) are the instantaneous current and voltage at time *t*, and *T* is the period of voltage. The underwater plasma-induced reactive species were identified using an optical emission spectrum (OES) signal (Figure 2b). Due to the low excitation threshold energy of 2.1 eV, the Na signal shows the highest intensity [28]. O atom, OH line, and H lines were present due to the plasma generation in water vapor.

The characteristics of plasma-treated water can be changed under the action of underwater plasma with a large number of reactions [29]. In this study, we evaluate the changes in water chemistry. With 30 min of plasma treatment, the concentration of hydrogen peroxide was increased to 17.21 mg L^−1^ as presented in Figure 3a. Figure 3b shows that the temperature was increased from 26 to 55 °C. Meanwhile, no difference in salinity and pH was found during 30 min of plasma treatment. There are several studies dealing with hydrogen peroxide generation during underwater plasma discharge [20]. This can be created from the reaction of two OH radicals through the following reaction (4):(4)$$\mathrm{OH}\cdot +\;\mathrm{OH}\cdot \to H_{2}O_{2}$$

With a high applied voltage, plasma current joule heating induces the evaporation of water resulting in fine bubbles [17]. These processes are causing an increase in solution temperature. As a high temperature (over 60 °C) is not sustainable due to potential damage to the plasma reactor and all lines, a maximum of 30 min was considered in this study [30].

### 3.3. Reduction of Microorganisms and Organic Matter in Waste Brine with the Underwater Plasma

Figure 4 illustrates the effects of underwater plasma treatment on the reduction of microorganisms in WB. Due to complexity of bacterial community in WB, its efficacy was evaluated using four different agars described in Section 2.4. The bacterial counts before treatment from WB were 5.87 ± 0.01, 6.09 ± 0.06, 5.71 ± 0.02, and 5.55 ± 0.05 log CFU mL^−1^ for being cultured on TSA, MA, MRS, and YPD agars, respectively. With a 30 min treatment time, the counts of the microorganisms were reduced to 0.5–2.51 log CFU mL^−1^. Based on the microbial populations after underwater plasma treatment, two inactivation kinetic models (log-linear model, Weibull model) were compared by calculating R^2^. Two models were fitted well, but the log-linear model (R^2^: 0.96–0.99) was slightly better than those of the Weibull model (Appendix A, Table 2). Our experiment revealed that underwater can reduce the number of all kinds of microorganisms, significantly indicating that the reuse of WB is feasible.

The influence of plasma treatment on the concentration of organic matters expressed as COD is shown in Figure 5. Initial COD was 320 ± 95.4 mg L^−1^, and 340 ± 43.6 mg L^−1^ of COD was measured after 30 min of underwater plasma discharge, indicating that no significant reduction was found in spite of the 30 min treatment (*p* > 0.05).

### 3.4. The Microbial Inactivation and Changes in Quality of Kimchi Cabbage Salted by PTWB

The reusability of each brine (NMB, WB, and PTWB) was evaluated with five parameters (salinity, pH, acidity, reducing sugar, and the population of microorganisms cultured on four different agars). Each parameter was measured immediately after desalination and after 7 days storage to ensure the safety of salted Kimchi cabbage [31]. It should be noted that the initial salinity of NMB (about 10%) was different from that of WB or PTWB (about 7%), so salts were added to WB and PTWB to be 10%.

#### 3.4.1. Microbial Inactivation of Kimchi Cabbage

Table 3 shows the counts of various microorganisms in salted Kimchi cabbage with desalination and after 7 days of storage. The Kimchi cabbage that was salted using NMB shows 5.89, 5.88, 5.82, and 5.94 log CFU g^−1^ by TSA, MRS, MA, YPD agar, respectively. Bacterial counts were much higher when WB was used as a salting material. Meanwhile, most of the bacterial population was not significantly different between the NMB- and PTWB-salted Kimchi cabbage. After 7 days, overall bacterial counts were decreased irrespective of salting solution. Nevertheless, the amount of reduction was different. A small reduction was observed in the WB-salted Kimchi cabbage: 0.77, 0.91, 0.85, and 0.59 log CFU g^−1^. In contrast, the microbial counts in NMB- and PTWB-salted Kimchi cabbage show a higher reduction. There is no significant difference in MRS-, MA-, and YPD-cultured microorganisms in NMB- and PTWB-salted Kimchi cabbage. Meanwhile, lower counts of total aerobic bacteria (cultured in TSA) were found in PTWB-salted Kimchi cabbage (*p* < 0.05).

#### 3.4.2. Salinity, pH, Acidity, and Reducing Sugar Content of Kimchi Cabbage

The changes in salinity, pH, and reducing sugar content of the salted Kimchi cabbage are shown in Table 4. The salinity of Kimchi cabbage prepared using NMB and PTWB is higher than that for WB. After 7 days, the salinity of all Kimchi cabbage decreased. There was significant difference in the following order: WB < NMB < PTWB. There were no differences in the pH values among all Kimchi cabbage after desalination and after 7 days of storage. The titratable acidities of Kimchi cabbage show no difference in Kimchi cabbage salted by NMB and PTWB. In contrast, the acidity of Kimchi cabbage salted by WB was low showing 0.144%. After 7 days, acidity was slightly increased to 0.17% irrespective of salting solution. There was a significant difference in the reducing sugar content of the NMB-, WB-, and PTWB-salted Kimchi cabbage. The reducing sugar content of WB used Kimchi cabbage shows the highest values, while PTWB shows the lowest values. The values were decreased slightly after 7 days, but their trend was not changed.

## 4. Discussion

During underwater plasma discharge, the generation of OH radical (A^2^Σ^+^-X^2^Π) at 309 nm which is one of the most powerful oxidants was observed. In addition, atomic oxygen with a high oxidation potential of 2.43 V was strongly detected in the plasma region [32]. As plasma is generated in water, these strong short-lived species are capable of reacting with other contaminants directly. With the 10-min treatment, a low concentration of hydrogen peroxide was found, indicating that most of the OH radicals were consumed to react with other contaminants. After that, the H_2_O_2_ concentration was increased since residual OH radical concentration is increased probably due to a low concentration of microorganisms. Nevertheless, the COD value did not change by plasma discharge. COD might be attributed to organic materials such as lactic acid, oxalic acid, acetic acid, and malic acid [33]. It is difficult to remove these organic materials. For instance, it takes more than 60 days to remove 20% of lactic acid in an aqueous solution kept at 80 °C [34]. In other research, photocatalytic treatment using a 500 W high-pressure mercury lamp was conducted in a 20 mL solution containing 10 mg TiO_2_. Only 3% of lactic acid was removed after 180 min of treatment [35]. In our experiment, no significant reduction was observed. It might seem that 30 min of underwater plasma treatment is not sufficient to remove organic material. Nevertheless, previous research shows that underwater treatment is capable to degrade various organic materials including phenol and azo dyes, implying that it is feasible to potentially remove organic materials in WB [36,37].

As expected, there are various microorganisms in waste brine. Most microorganisms are known as non-pathogens. However, some bacteria such as *Serratia marcescens* found in YPD agar has been reported to cause an outbreak of *Serratia marcescens* infections in neonates [20]. Thus, bacterial inactivation should be conducted to reuse waste brine. Despite the presence of organic matter, our results clearly show that underwater plasma treatment is an effective method for the removal of microorganisms cultured in four different agars. A 3.3–5.2 log reduction was observed during the 30 min treatment to examine the sensitivity of foodborne pathogens to underwater plasma treatment based on the survival curve and two different fitting models. The log-linear model fitted the data better than the Weibull model for the inactivation kinetics of microorganisms irrespective of kind of agar. Many studies have been conducted on the behavior of microorganisms in the aqueous phase by plasma treatment. The Weibull model is frequently used to predict microbial inactivation by cold plasma or plasma-activated water treatment [38,39,40]. In contrast, the *B. cereus* spores inactivation curve by air-fed plasma jet was adequately fitted by the log-logistic model [41]. In general, there are numerous post-discharge reactions in plasma-activated water. Some reactive species such as ^1^O_2_, OH·, and peroxynitrite produced by complex chemical reactions are involved in the bactericidal effect. In this study, the log-linear model shows a good fit to predict microbial inactivation, indicating that underwater plasma generates continuous production of reactive species including OH and atomic O, and they are directly reacting with microorganisms.

PTWB was used in the Kimchi cabbage salting process and the results were compared to NMB- and WB-salted Kimchi cabbage. In general, salted Kimchi cabbage was used after desalination with tap water, and the desalinated Kimchi cabbage that was stored in 4 °C for 7 days was tested. Due to the presence of numerous of microorganisms, a much higher population of bacteria was found in WB-salted Kimchi cabbage. In contrast, no significant difference was found between NMB- and PTWB-salted Kimchi cabbage. The counts of TSA-, MRS-, MA-, and YPD-cultured microorganisms were reduced during 7 days of fermentation. Nevertheless, WB-salted Kimchi cabbage shows the highest population of microorganisms. Although PTWB-salted Kimchi cabbage showed similar counts in microorganisms cultured in MRS, MA, and YPD, fewer counts were found in TSA agar. It is probably attributed to the further bactericidal effect due to the residual hydrogen peroxide. For instance, we have previously discovered a significant reduction in a representative bacteria, *Escherichia coli*, after 5 days of reaction with underwater plasma-treated brine [20]. Nevertheless, the greater reduction in TSA-cultured microorganisms compared to other media should be studied further.

The reduction of salinity and reducing sugar during 7 days of Kimchi cabbage fermentation is presented in Table 4. To manufacture salted Kimchi cabbage, there are some variables including the concentration of salt (8–30%), salting time (8–18 h), and temperature (10–40 °C) [42,43,44], and the best salinity was 1.89–3.36% [45]. In this study, the salinity of NMB- and PTWB-salted Kimchi cabbage shows 1.96 and 1.95%. respectively, indicating that the use of PTWB is suitable in the recycling process. Increasing the storage period, however, is not recommended since all Kimchi cabbage did not meet the salinity criteria. The reducing sugar of the NMB-salted Kimchi cabbage was 414.95 mg L^−1^ which is higher than the PTWB-salted Kimchi cabbage at 398 mg L^−1^. Similar results have been presented for ozonated Kimchi cabbage [6]. The contents of the reducing sugar influence in the fermentation period do not significantly change the quality of the Kimchi cabbage [46]. The pH of NMB-salted Kimchi cabbage was 6.2 and it was decreased to 6.0 after 7-days storage. Reducing pH can be derived from the formation of organic acid lactic acid bacteria [47]. A previous study reported similar results showing that pH value was maintained at around 6.0 at the first 2 weeks of storage [48]. The acidity of NMB-salted Kimchi cabbage was 0.161% which is lower than that reported by Song et al., but higher than the acidity reported by Park et al. [47,48]. It is attributed to the different salting method, and the concentration of salt [47]. Nevertheless, no significant difference in pH and acidity was found in NMB- and PTWB-salted Kimchi cabbage. Our study shows that underwater plasma treatment is potentially feasible to reuse the waste brine salting process of Kimchi production.

## 5. Conclusions

This is the first report on the reuse of spent brine with underwater plasma treatment to inactivate foodborne pathogens in WB and salted Kimchi cabbage. Despite little changes in COD values, treatment of WB with underwater plasma is effective in reducing bacterial populations. The efficacy of treatment depends on the duration of exposure, and 30 min of the treatment shows approximately 3.36, 3.9, 5.21, and 3.48 log CFU mL^−1^ reductions in microorganisms cultured on TSA, MA, MRS, and YPD agars, respectively. The log-linear model was fitted well with high R^2^ values to predict microbial reduction during plasma treatment. The microbial population of Kimchi cabbage salted by WB was 6.5–6.83 log CFU g^−1^ depending on the culturing medium, which was much higher than that salted by NMB or PTWB. Meanwhile, there was no significant difference in the bacterial concentration in Kimchi cabbage salted by NMB and PTWB. Furthermore, the quality of Kimchi cabbage treated with PTWB shows significantly higher salinity than that treated with WB. Overall, the use of underwater plasma is a promising technology for the microbiological safety of salted food since it is a non-thermal process. Further studies are required to examine the feasibility of long-term storage of Kimchi cabbage salted by PTWB, to allow for the application of underwater plasma treatment in the real Kimchi production industry.

## Figures and Tables

**Figure 1 foods-12-00728-f001:**
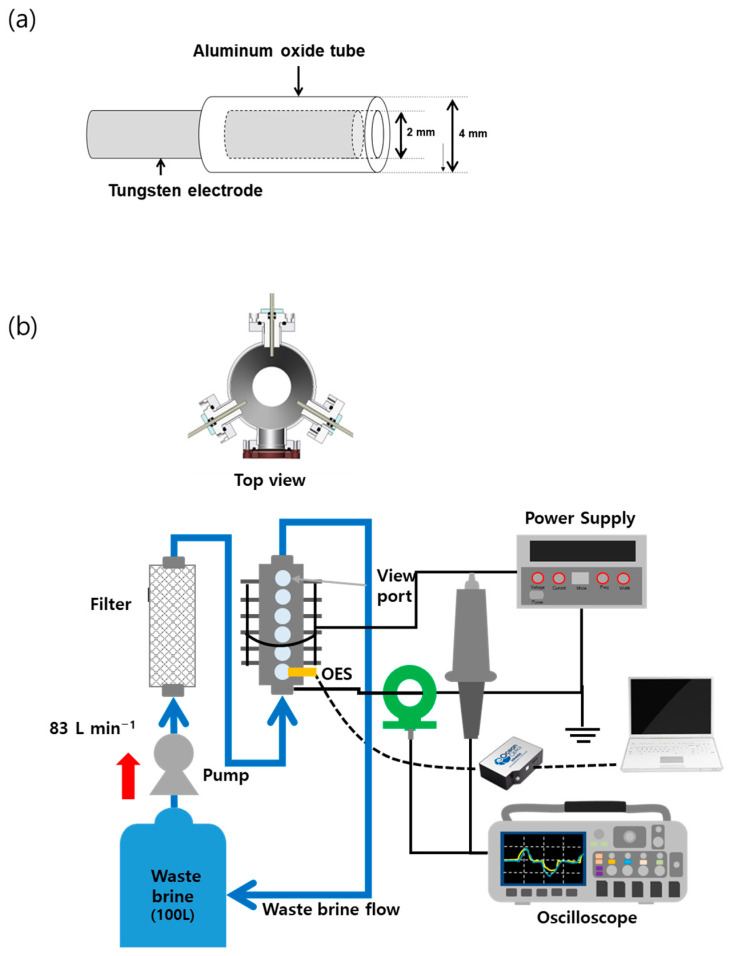
Experimental setup for the underwater plasma discharge device. (**a**) The shape of the capillary electrode; (**b**) configuration of the treatment system.

**Figure 2 foods-12-00728-f002:**
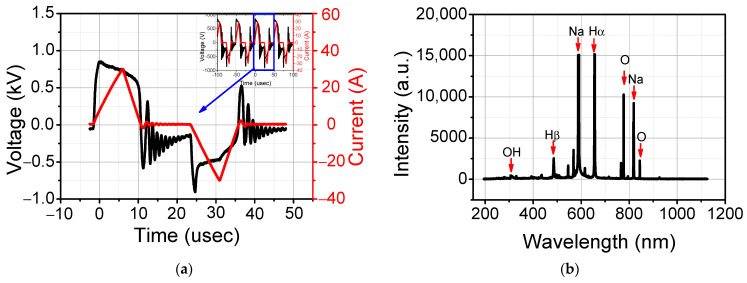
The characteristics of underwater plasma in saline water. (**a**) Voltage and current signals; (**b**) optical emission spectrum (OES) signals.

**Figure 3 foods-12-00728-f003:**
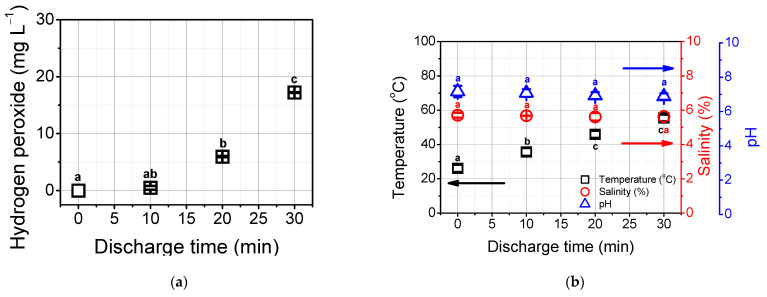
The characteristics of underwater PTWB. (**a**) The concentration of hydrogen peroxide; (**b**) the changes in temperature, pH, and salinity. Different letters indicate significant differences (*p* < 0.05).

**Figure 4 foods-12-00728-f004:**
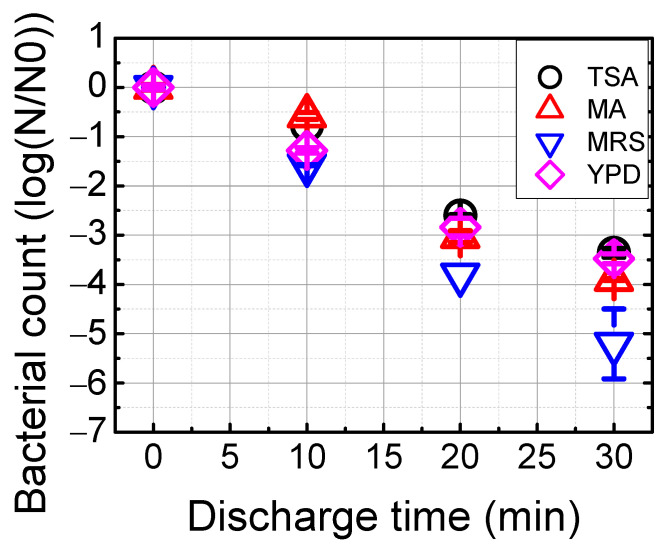
The effect of underwater plasma treatment on the population of microorganisms in WB.

**Figure 5 foods-12-00728-f005:**
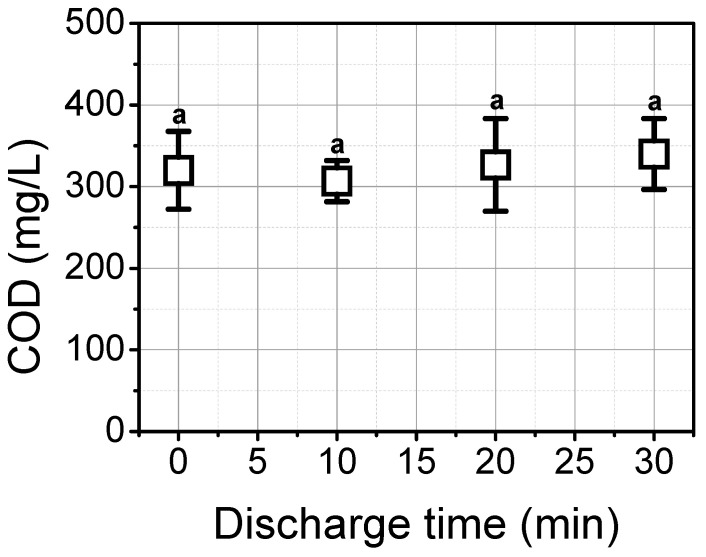
The effect of underwater plasma treatment on the chemical oxygen demand (COD) in waste brine. Different letters indicate significant differences (*p* < 0.05).

**Table 1 foods-12-00728-t001:** Characteristics of WB from the salting process of Kimchi cabbage.

pH	Conductivity (mS cm^−1^)	Salinity (%)	COD (mg L^−1^)
5.71 ± 0.14	100.7 ± 8.71	7.00 ± 0.33	320 ± 95.4
TSA (CFU mL^−1^)	MA (CFU mL^−1^)	MRS (CFU mL^−1^)	YPD (CFU mL^−1^)
5.87 ± 0.01	6.09 ± 0.06	5.71 ± 0.02	5.55 ± 0.05

Abbreviations: TSA: Tryptic Soy Agar, MA: Marine agar, MRS: de Man Rogosa Sharpe Agar, YPD: Yeast Extract–Peptone–Dextrose.

**Table 2 foods-12-00728-t002:** Evaluation of the log-linear model and Weibull model to estimate the reduction of microorganisms in WB by underwater plasma treatment.

Model	Agar	Parameter		R^2^
Log-linear	TSA	k	−0.115	0.983
	MA	k	−0.131	0.962
	MRS	k	−0.177	0.996
	YPD	k	−0.124	0.989
Weibull	TSA	α	4.47	0.954
		β	1.098	
	MA	α	4.886	0.921
		β	1.238	
	MRS	α	2.447	0.989
		β	1.000	
	YPD	α	2.314	0.977
		β	0.825	

Abbreviations: TSA: Tryptic Soy Agar, MA: Marine agar, MRS: de Man Rogosa Sharpe Agar, YPD: Yeast Extract–Peptone–Dextrose.

**Table 3 foods-12-00728-t003:** The changes in the microbial population in salted Kimchi cabbage.

	After 0 Day				After 7 Days			
	TSA	MRS	MA	YPD	TSA	MRS	MA	YPD
NMB	5.89 ± 0.18 ^a^	5.88 ± 0.18 ^a^	5.82 ± 0.03 ^a^	5.94 ± 0.28 ^a^	5.34 ± 0.04 ^b^	4.30 ± 0.25 ^a^	4.53 ± 0.07 ^a^	4.62 ± 0.21 ^a^
WB	6.75 ± 0.04 ^b^	6.83 ± 0.01 ^b^	6.50 ± 0.04 ^b^	6.60 ± 0.06 ^b^	5.98 ± 0.02 ^c^	5.92 ± 0.05 ^b^	5.65 ± 0.03 ^b^	6.01 ± 0.04 ^b^
PTWB	6.07 ± 0.05 ^a^	6.12 ± 0.3 ^a^	5.89 ± 0.07 ^a^	5.65 ± 0.72 ^a^	4.87 ± 0.20 ^a^	4.65 ± 0.27 ^a^	4.59 ± 0.25 ^a^	4.45 ± 0.02 ^a^

Abbreviations: TSA: Tryptic Soy Agar, MA: Marine agar, MRS: de Man Rogosa Sharpe Agar, YPD: Yeast Extract–Peptone–Dextrose. Different letters in the same column indicate significant differences (*p* < 0.05).

**Table 4 foods-12-00728-t004:** The changes in 4 quality parameters of salted Kimchi cabbage.

	After 0 Day				After 7 Days			
	Salinity (%)	pH	Acidity (%)	RS (mg L^−1^)	Salinity (%)	pH	Acidity (%)	RS (mg L^−1^)
NMB	1.96 ± 0.027 ^b^	6.2 ± 0.01 ^a^	0.161 ± 0.0019 ^b^	414.95 ± 4.46 ^b^	1.62 ± 0.023 ^b^	6.0 ± 0.01 ^a^	0.17 ± 0.0021 ^a^	409.79 ± 3.63 ^b^
WB	1.69 ± 0.023 ^a^	6.2 ± 0.01 ^a^	0.144 ± 0.0014 ^a^	428.86 ± 2.16 ^c^	1.51 ± 0.014 ^a^	5.9 ± 0.03 ^a^	0.17 ± 0.0014 ^a^	418.21 ± 3.06 ^c^
PTWB	1.95 ± 0.047 ^b^	6.1 ± 0.03 ^a^	0.163 ± 0.0056 ^b^	398.00 ± 3.98 ^a^	1.77 ± 0.014 ^c^	5.9 ± 0.02 ^a^	0.17 ± 0.0036 ^a^	353.25 ± 1.36 ^a^

Abbreviations: TSA: Tryptic Soy Agar, MA: Marine agar, MRS: de Man Rogosa Sharpe Agar, YPD: Yeast Extract–Peptone–Dextrose, RS: reducing sugar content. Different letters in the same column indicate significant differences (*p* < 0.05).

## Data Availability

Data is contained within the article and Appendix A.

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
