# Peer review of "Bactericidal Effect of Underwater Plasma Treatment on Waste Brine from Kimchi Production Process and the Evaluation of Reusability of Plasma-Treated Waste Brine in Salting Kimchi Cabbage"

_foods, 2023, doi:10.3390/foods12040728_

Round 1

Reviewer 1 Report

  I have reviewed the manuscript entitled ‘Bactericidal Effect of Underwater Plasma Treatment on Waste Brine from Kimchi Production Process for Recycling Waste Saline ’. There are several issues should the authors be considered:

1. This paper focuses on the effects of plasma on microorganisms in waste brine and the effects of plasma-treated waste brine on the quality of pickles. I think the topic should be slightly modified because it contains only one aspect.

2. The plasma treatment time selected in this paper is 30 minutes. Is it the best treatment time? Or the longer the treatment time is beneficial to the recycling of waste brine.

3. If the curves of two bacterial inactivation models are shown will it be more intuitive?

4. Line 212.Please check whether it is 16s rRNA or 16s rDNA?

5. What is the reason why chemical oxygen demand value not only does not decrease but also increases slightly during pickling?

6. Line 287.Please explain in Part 4 why the number of bacteria decreases after seven days of desalination

7. Please add standard deviation to the data in Table 1.

8. The discussion should be enlarged.

Author Response

Dear Reviewer 1:

[January 17, 2023]

I am pleased to submit an article entitled “Bactericidal Effect of Underwater Plasma Treatment on Waste Brine from Kimchi Production Process and the Evaluation of reusability of Plasma-Treated Waste Brine in Salting Kimchi Cabbage” for consideration for publication in Foods.

In this manuscript, we showed that underwater plasma treatment was effective in the reduction of microorganisms cultured in Trypticase Soy agar (TSA), Marine agar (MA), de Man Rogosa Sharpe Agar (MRS), and Yeast Extract–Peptone–Dextrose (YPD). Their inactivation behavior fitted well with the log-linear model. In addition, there was no significant difference in the quality of kimchi cabbage salted by plasma-treated waste saline and newly made saline, indicating that the use of underwater plasma treatment can be considered as an alternative approach to ozone and conventional chlorine-based sanitizers. We believe our findings would appeal to the readership of applied physics and the food industry.

We received 8 comments from professional reviewer 1. and we try to improve the quality of the paper by editing the original paper. Each reviewer’s comments and advice, and responses about them were reported in the attached file. We hope our paper is helpful to other researchers who wants to study similar area.

Yours sincerely,

Junghyun Lim

Korea Institute of Fusion Energy

37 Dongjangsan-ro

Gunsansi, Jeollabuk-do, Republic of Korea

Tel: +82-10-4479-4791

E‐mail: [email protected]

Reviewer 2 Report

The manuscript is well written and is scientifically sound to reuse and decontaminate saline water using cold plasma novel technology. I have provided some comments in the attached PDF. Authors should especially check uniformity in abbreviations units used throughout the manuscript. 

Author Response

Dear Review 2:

[January 17, 2023]

I am pleased to submit an article entitled “Bactericidal Effect of Underwater Plasma Treatment on Waste Brine from Kimchi Production Process and the Evaluation of reusability of Plasma-Treated Waste Brine in Salting Kimchi Cabbage” for consideration for publication in Foods.

In this manuscript, we showed that underwater plasma treatment was effective in the reduction of microorganisms cultured in Trypticase Soy agar (TSA), Marine agar (MA), de Man Rogosa Sharpe Agar (MRS), and Yeast Extract–Peptone–Dextrose (YPD). Their inactivation behavior fitted well with the log-linear model. In addition, there was no significant difference in the quality of kimchi cabbage salted by plasma-treated waste saline and newly made saline, indicating that the use of underwater plasma treatment can be considered as an alternative approach to ozone and conventional chlorine-based sanitizers. We believe our findings would appeal to the readership of applied physics and the food industry.

We received 6 reviews from professional reviewer 2. and we try to improve the quality of the paper by editing the original paper. Each reviewer’s comments and advice, and responses about them were reported in the attached file. We hope our paper is helpful to other researchers who wants to study similar area.

Yours sincerely,

Junghyun Lim

Korea Institute of Fusion Energy

37 Dongjangsan-ro

Gunsansi, Jeollabuk-do, Republic of Korea

Tel: +82-10-4479-4791

E‐mail: [email protected]

Reviewer 3 Report

Dear Author, I reviewed the manuscript (foods-2169748) entitled Bactericidal Effect of Underwater Plasma Treatment on Waste Brine from Kimchi Production Process for Recycling Waste Saline. This manuscript presents relevant information about underwater plasma treatments to reduce microorganisms of Kimchi waste brine. However, some sections of the presented data can be improved. For this reason, I consider that this manuscript needs changes to be considered.

Additional comments.

Highlight the advantages of underwater plasma treatments to reduce microorganism counts in water for the food industry.

Check paragraph extension in this manuscript.

Try to detail how the antibacterial activity was detected in water samples. 

Include an experimental design containing statistical factors and response variables in the statistical analyses applied to the findings of this research.

Include future trends to keep working with the obtained data. 

Try to conclude with a general statement of the most relevant part of this study.

Author Response

Dear Review 3:

[January 17, 2023]

I am pleased to submit an article entitled “Bactericidal Effect of Underwater Plasma Treatment on Waste Brine from Kimchi Production Process and the Evaluation of reusability of Plasma-Treated Waste Brine in Salting Kimchi Cabbage” for consideration for publication in Foods.

In this manuscript, we showed that underwater plasma treatment was effective in the reduction of microorganisms cultured in Trypticase Soy agar (TSA), Marine agar (MA), de Man Rogosa Sharpe Agar (MRS), and Yeast Extract–Peptone–Dextrose (YPD). Their inactivation behavior fitted well with the log-linear model. In addition, there was no significant difference in the quality of kimchi cabbage salted by plasma-treated waste saline and newly made saline, indicating that the use of underwater plasma treatment can be considered as an alternative approach to ozone and conventional chlorine-based sanitizers. We believe our findings would appeal to the readership of applied physics and the food industry.

We received 5 reviews from professional reviewer 3. and we try to improve the quality of the paper by editing the original paper. Each reviewer’s comments and advice, and responses about them were reported in the attached file. We hope our paper is helpful to other researchers who wants to study similar area.

Yours sincerely,

Junghyun Lim

Korea Institute of Fusion Energy

37 Dongjangsan-ro

Gunsansi, Jeollabuk-do, Republic of Korea

Tel: +82-10-4479-4791

E‐mail: [email protected]

Round 2

Reviewer 1 Report

  The new version of the manuscript has been revised following the request. There are several minor issues should the authors be considered. In my opinion, it can be accept after minor revised.

 1.In this paper, four processing times of 0,10,20 and 30 minutes are selected. The results show that the effect is the best after 30 minutes of processing. Is the longer the processing time the better ?

2. Line 17. the word was should be deleted.

3. Line 81.This sentence needs to be reorganized. The word research is suspected to be repetitive.

Author Response

Response to Reviewer 1 Comments

Point 1: In this paper, four processing times of 0,10,20, and 30 minutes are selected. The results show that the effect is the best after 30 minutes of processing. Is the longer the processing time the better?

Response 1: We think 30 minutes is the best treatment time. Of course, more treatment time results in more inactivation efficacy. Nevertheless, 30 min of treatment time is better as too long treatment time results in high temperature that might have a negative effect on the reactor. Moreover, waste brine treated with plasma for 30 minutes shows similar results compared with newly made brine as presented in table 3, implying that more treatment time is not good economically.

Before edit:

As thermal inactivation can be considered during underwater plasma treatment,

After edit:

As a high temperature (over 60 ℃) is not sustainable due to potential damage to the plasma reactor and all lines, a maximum of 30 minutes was considered in this study [30]

Point 2: Line 17. the word “was” should be deleted.

Response 2: Thank you for your recommendation. Nevertheless, I think the word “was” is necessary to express what I want to say. Microbial inactivation was measured using 4 different agars. Thus, the word “was” should be used in this sentence.

The inactivation efficacy was analyzed using 4 different agars (Tryptic Soy Agar (TSA), Marine agar (MA), de Man Rogosa Sharpe Agar (MRS), Yeast Extract-Peptone-Dextrose (YPD), respectively.

Point 3: Line 81. This sentence needs to be reorganized. The word “research” is suspected to be repetitive.

Response 3: As pointed out, we had better edit the word “research”

Before edit:

Our research group has investigated capillary discharge, one of the ways to generate underwater plasma, to eliminate pollutants for the last decade

After edit:

We have investigated capillary discharge, one of the ways to generate underwater plasma, to eliminate pollutants for the last decade
